# Cyclophilin Inhibition Protects Against Experimental Acute Kidney Injury and Renal Interstitial Fibrosis

**DOI:** 10.3390/ijms22010271

**Published:** 2020-12-29

**Authors:** Khai Gene Leong, Elyce Ozols, John Kanellis, Shawn S. Badal, John T. Liles, David J. Nikolic-Paterson, Frank Y. Ma

**Affiliations:** 1Monash Medical Centre, Department of Nephrology, Clayton, VIC 3168, Australia; khaigeneleong@gmail.com (K.G.L.); elyce.ozols@monash.edu (E.O.); john.kanellis@monash.edu (J.K.); frank.ma@monash.edu (F.Y.M.); 2Centre for Inflammatory Diseases, Monash University, Clayton, VIC 3168, Australia; 3Gilead Sciences, Foster City, CA 94404, USA; shawn.badal@gilead.com (S.S.B.); john.liles@gilead.com (J.T.L.)

**Keywords:** cyclophilin, acute kidney injury, cell death, chronic kidney disease, inflammation, macrophage, neutrophil, renal fibrosis

## Abstract

Cyclophilins have important homeostatic roles, but following tissue injury, cyclophilin A (CypA) can promote leukocyte recruitment and inflammation, while CypD can facilitate mitochondrial-dependent cell death. This study investigated the therapeutic potential of a selective cyclophilin inhibitor (GS-642362), which does not block calcineurin function, in mouse models of tubular cell necrosis and renal fibrosis. Mice underwent bilateral renal ischemia/reperfusion injury (IRI) and were killed 24 h later: treatment with 10 or 30 mg/kg/BID GS-642362 (or vehicle) began 1 h before surgery. In the second model, mice underwent unilateral ureteric obstruction (UUO) surgery and were killed 7 days later; treatment with 10 or 30 mg/kg/BID GS-642362 (or vehicle) began 1 h before surgery. GS-642362 treatment gave a profound and dose-dependent protection from acute renal failure in the IRI model. This protection was associated with reduced tubular cell death, including a dramatic reduction in neutrophil infiltration. In the UUO model, GS-642362 treatment significantly reduced tubular cell death, macrophage infiltration, and renal fibrosis. This protective effect was independent of the upregulation of IL-2 and activation of the stress-activated protein kinases (p38 and JNK). In conclusion, GS-642362 was effective in suppressing both acute kidney injury and renal fibrosis. These findings support further investigation of cyclophilin blockade in other types of acute and chronic kidney disease.

## 1. Introduction

Cyclophilins (Cyp) are ubiquitously expressed proteins, which belong to the immunophilin family [1,2]. All Cyp possess peptidyl-prolyl isomerase (PPIase) activity that catalyzes the interconversion of *cis* and *trans* isomers of proline that is important for correct protein folding [2,3,4]. Of the 18 members of the Cyp family, the two well-studied forms are Cyclophilin A (CypA) and Cyclophilin D (CypD).

Cyclophilin A (CypA) is a highly abundant cytosolic protein. During pathological conditions, CypA can be released in response to specific stimuli or released from damaged cells and act as a damage-associated molecular pattern (DAMP) to promote inflammation [1,3,5]. Extracellular CypA acts as a chemotactic factor for leukocytes such as neutrophils, monocytes and T cells [1]. In addition, CypA can act as a cytokine, via the cell-surface CD147 receptor, to induce activation of several pro-inflammatory signaling pathways [1,3,6,7]. Indeed, mice lacking the CypA gene are protected in models of acute myocardial infarction and autoimmune myocardial injury and fibrosis [8,9].

Cyclophilin D (CypD) is a soluble mitochondrial matrix protein that has a physiological role in folding of mitochondrial matrix proteins and in maintaining the Ca^2+^ balance between the endoplasmic reticulum and mitochondria [10,11,12]. Under pathological conditions, CypD has been implicated in promoting cell death via the opening of the mitochondrial permeability transition pore [10,13,14]. Furthermore, mice lacking the CypD gene are protected from cardiac and renal ischemia/reperfusion injury [13,15].

The immunosuppressive drug, cyclosporine A (CsA), binds to members of the cyclophilin family [16]. It is the ability of the CsA/CypA complex to bind to calcineurin that provides the ability of CsA to inhibit IL-2 production and T cell activation, which made it a first-line therapy in organ transplantation [17]. CsA also binds to and inhibits the action of CypD to prevent necrotic cell death [14]. However, a major limitation in the clinical use of CsA is the many undesired side effects of calcineurin inhibition, such as infections, nephrotoxicity, neurotoxicity, and malignancy risk [18,19]. For example, CsA can protect against experimental renal ischemia/reperfusion injury (IRI) by inhibiting CypD-mediated cell death [20], but its ability to reduce renal blood flow and increase renal vascular resistance can also exacerbate this type of injury [21], and long-term exposure to CsA induces tubular atrophy and interstitial fibrosis [19,22]. Indeed, great care is taken to monitor CsA levels in transplant patients to avoid inducing organ fibrosis. Thus, to investigate the therapeutic potential of selective cyclophilin inhibition, new cyclophilin inhibitors have been developed, which are either modified versions of the CsA cyclic peptide, or novel chemical structures, that do not impact upon calcineurin function [3].

We lack specific therapies to prevent, or treat, acute kidney injury whether induced by ischemic/reperfusion injury (IRI) or by other renal insults. This is not only an important issue in the short-term, but acute kidney injury can also lead to the development of chronic kidney disease, or exacerbate pre-existing kidney disease [23,24,25]. In addition, we have no specific therapies to suppress renal interstitial fibrosis—a common pathologic mechanism in the progression of all forms of chronic kidney disease to end-stage renal failure that requires treatment by dialysis or transplantation.

The study investigated a new cyclophilin inhibitor, GS-642362, derived from the naturally occurring sanglifehrin macrocycle which is a potent inhibitor of CypA [26]. GS-642362 has high bioavailability and inhibits CypA, CypB and CypD with IC50 values of 7, 18 and 21 nM, respectively [26]. We investigated whether GS-642362 can protect against tubular cell necrosis and loss of renal function induced by renal ischemia/reperfusion injury (IRI) and protect against cell death, inflammation and renal fibrosis induced by unilateral ureteric obstruction (UUO). 

## 2. Results

### 2.1. GS-642362 Prevents Oxygen Radical Induced Tubular Cell Death

Primary kidney tubular epithelial cells lacking the CypD gene show a significant, but not complete, protection against H_2_O_2_-induced cell death [27]. Using this assay, GS-642362 showed a dose-dependent inhibition of H_2_O_2_-induced cell death in primary mouse tubular epithelial cells from wild type mice (Figure 1).

### 2.2. GS-642362 Protects Against Renal Ischemia/Reperfusion Injury (IRI)

We examined whether prophylactic treatment with GS-642362 could prevent acute kidney injury in a model of severe renal IRI. Both CypA and CypD are highly expressed in the kidney. The induction of renal IRI did not affect CypA or CypD mRNA levels (Figure 2A,B). Vehicle-treated mice exhibited a severe loss of renal function at 24 h after IRI, as indicated by a 15-fold increase in plasma creatinine levels (Figure 2C), severe tubular necrosis in the outer medulla and inner cortex of the kidney (Figure 2D,E), and a marked upregulation of the tubular damage marker, neutrophil gelatinase-associated lipocalin (NGAL/LCN2) (Figure 2I). Many dying tubular cells were also detected by TUNEL staining in vehicle-treated mice (Figure 3A,B), and a heavy neutrophil infiltrate was evident in the area of damaged tubules (Figure 4A,B). There was also a macrophage infiltrate as indicated by an increase in CD68 mRNA levels and increased mRNA levels of the monocyte chemoattractant CCL2, the pro-inflammatory cytokine, TNF, and IL-2 (Figure 4E–H).

As expected, GS-642362 did not affect mRNA levels of CypA or CypD (Figure 2A,B). However, GS-642362 treatment gave profound and dose-dependent protection against the loss of renal function in renal IRI, with 54 and 85% improvements in plasma creatinine levels with 10 and 30 mg/kg doses, respectively (Figure 2C). The 30 mg/kg dose of GS-642362 also provided substantial protection against histologic tubular damage (Figure 2G,H), and both doses of GS-642362 significantly reduced mRNA levels of NGAL/LCN2 (Figure 2I). In addition, both doses of GS-642362 significantly reduced the number of TUNEL+ dying cells (Figure 3C,D). Both doses of GS-64362 caused a dramatic reduction in neutrophil infiltration (Figure 4C,D), while there was also a reduction in macrophage infiltration based on CD68 mRNA levels, which was associated with a reduction in CCL2 mRNA levels and a reduction in TNF mRNA levels (Figure 4E–G). However, the upregulation of IL-2 mRNA levels was not affected by GS-642362 treatment (Figure 4H). Peak plasma levels of GS-642362 were 1.95 ± 0.78 and 4.72 ± 2.48 µmol/L for the 10 mg and 30 mg/kg/BID doses, respectively, measured at the time of killing, which was 1 h after the last oral gavage.

### 2.3. GS-642362 Reduces Fibrosis in the Unilateral Ureteric Obstruction (UUO) Model

We next examined whether GS-642362 treatment could prevent the development of inflammation and fibrosis in the UUO model. There was no change in the mRNA levels of CypA or CypD in the UUO model compared to normal mice (Figure 5A,B). Vehicle-treated UUO mice exhibited marked tubular dilation compared to normal mice (Figure 5C,D,F), which was associated with increased mRNA levels of the tubular damage marker, NGAL/LCN2 (Figure 5G). However, tubular damage is much less aggressive in the UUO model compared to renal IRI, with no detectable tubular necrosis. Compared to normal mouse kidney, untreated and vehicle-treated day 7 UUO groups exhibited a significant increase in the number of TUNEL+ dying cells, which were predominantly tubular cells (Figure 6A,B,D). Vehicle and untreated mice also showed a substantial macrophage infiltrate based upon F4/80 immunostaining and CD68 mRNA levels (Figure 7A,B,D,E). This was associated with upregulation of monocyte chemokine CCL2 (Figure 7F). There was also upregulation of IL-2 mRNA levels (Figure 7G).

GS-642362 treatment did not affect CypA or CypD mRNA levels in the UUO kidney (Figure 5A,B). In addition, GS-642362 treatment did not affect the marked tubular dilation, the degree of tubular histologic damage, or upregulation of NGAL/LCN2 mRNA levels (Figure 5E–G). However, GS-642362 treatment substantially reduced the number of TUNEL+ cells in the day 7 UUO kidney (Figure 6C,D) and caused a small, but significant, reduction in the macrophage infiltrate and in CD68 and CCL2 mRNA levels (Figure 7C–H). However, GS-642362 treatment did not affect upregulation of IL-2 mRNA levels (Figure 7G). Peak plasma levels of GS-642362 were 4.01 ± 2.27 and 6.1 ± 0.8 µmol/L for the 10 mg and 30 mg/kg/BID doses, respectively, measured at the time of killing, which was 1 h after the last oral gavage.

Vehicle and untreated day 7 UUO groups exhibited substantial renal fibrosis based upon an increase in the total kidney collagen content, as measured by the hydroxyproline assay (Figure 8A), an increase in the interstitial deposition of collagen IV (Figure 8B,C,E), and interstitial accumulation of α-SMA+ myofibroblasts (Figure 8F). In addition, mRNA levels of pro-fibrotic molecules (collagen IV, α-SMA, TGF-β1, and PDGF-B) were upregulated in vehicle and untreated day 7 UUO kidney (Figure 8G–J). There was also a significant loss of peritubular capillaries based upon CD31 immunostaining in vehicle and untreated day 7 UUO groups (Figure 9A,B,D). Finally, there was a clear increase in the activation (phosphorylation) of the p38 mitogen-activated protein kinase (MAPK) and the JUN amino terminal kinase (JNK) in the day 7 UUO kidney (Figure 10A–D).

GS-642362 treatment clearly reduced the total collagen content in the day 7 UUO kidney (Figure 8A). Consistent with this finding, GS-642362 treatment significantly reduced interstitial collagen IV deposition (Figure 8D,E) and interstitial α-SMA+ myofibroblast accumulation (Figure 8F). GS-642362 treatment also significantly reduced mRNA levels of collagen IV, α-SMA, and TGF-β1 but did not affect PDGF-B mRNA levels (Figure 8G–J). GS-642362 treatment gave a non-significant trend towards protection against the loss of peritubular capillaries (Figure 9C,D) but had no effect on the activation of p38 MAPK and JNK enzymes (Figure 10A–D).

## 3. Discussion

Our studies identified that cyclophilin inhibition with GS-642362 provides protection against severe acute kidney injury in the IRI model and provided partial, but significant, protection against renal fibrosis in the UUO model. These are important findings, since there are currently no specific therapies to prevent or treat acute kidney injury or progressive renal fibrosis.

Prophylactic treatment with GS-642362 gave a dose-dependent protection against acute kidney injury in the renal IRI model; however, GS-642362 inhibition of cell death, macrophage infiltration, and fibrosis in the UUO model failed to show a similar dose dependency. In the IRI model, there was a 2.4-fold difference in peak plasma levels of GS-642362 and which approximated to EC_50_ and EC_90_ target coverage for the 10 and 30 mg/kg doses, respectively. Analysis of peak drug levels on day 7 in the UUO model showed an accumulation of circulating GS-642362 over time, with a 2-fold increase in plasma GS-642362 levels with the 10 mg/kg dose, whereas a much lesser increase (29%) was evident with the 30 mg/kg dose. The higher plasma levels of GS-642362 in the 10 mg/kg treatment group indicates that this dose was approaching EC_90_ target coverage, which may explain the lack of dose response seen in most of the parameters in the UUO model. Of note, GS-642362 did not affect the upregulation of IL-2 mRNA levels in either the IRI or UUO model, demonstrating that it does not inhibit NFAT signaling. In addition, we confirmed that GS-642362 did not affect mRNA levels of CypA or CypD in these models.

GS-642362 gave very substantial protection against loss of kidney function and a significant reduction in tubular cell death, as shown by reduced tubular necrosis and TUNEL+ cells, in a model of severe IRI. This is consistent with the ability of GS-642362 to inhibit CypD. Indeed, GS-642362 inhibition of H_2_O_2_-induced tubular cell death mirrored the protection of CypD−/− tubular cells in the same assay [27]. Previous studies have shown that CypD−/− mice show significant protection against acute renal failure and tubular necrosis in models of varying severity of IRI [15,28,29]. Of these studies, Linkermann et al. [28] used a model of severe renal ischemia, which is most comparable to our experiments. Although not a direct comparison, we found an 85% reduction in serum creatinine levels at 24 h after IRI with GS-642362 treatment, whereas CypD−/− mice showed an approximately 35% reduction in serum creatinine compared to wild type mice at 48 h after IRI [28], suggesting that GS-642362 may provide protection beyond simply inhibiting CypD.

GS-642362 treatment dramatically reduced neutrophil infiltration into the kidney following IRI. This may reflect a reduction in the release of DAMPs as a result of reduced tubular cell damage, thereby reducing the recruitment and activation of neutrophils, which contribute to a “second wave” of tubular cell death in models of renal IRI [30,31,32]. Extracellular CypB can induce the chemotaxis of inflammatory cells into damaged tissues [33], although no studies of CypB−/− mice in IRI models have been reported. CypA is released into the extracellular environment early in the necroptosis pathway [34], and CypA is recognized as a DAMP and a neutrophil chemotactic factor [1,5,35]. CypA−/− mice were protected against loss of renal function and tubular cell death in a 24 h model of renal IRI model. This protection was attributed to the profound reduction in neutrophil infiltration in CypA−/− mice, given that CypA−/− tubular cells are not protected from reactive oxygen species-induced cell death [36]. Thus, the dramatic protection seen against the loss of renal function with GS-642362 treatment in severe renal IRI may be due to the additive benefits of blockade of both CypD-mediated tubular necrosis and inhibition of CypA-dependent neutrophil-mediated tubular cell death and damage. However, formal demonstration that GS-642362 provides greater protection than that seen with blockade of CypA or CypD alone requires further experiments, which are beyond the scope of these studies.

Surgical UUO induces a rapid and aggressive renal interstitial fibrosis in association with tubular cell death and macrophage inflammation [37,38]. One limitation of this model is that retaining one normal kidney means that fibrosis in the UUO kidney cannot be related to renal function or proteinuria, key clinical measures in chronic kidney disease; however, UUO does provide a highly reproducible and rapidly progressive model in which to investigate mechanisms of renal fibrosis. GS-642362 treatment significantly reduced tubular cell death, macrophage infiltration, and renal fibrosis on day 7 of the UUO model, arguing that cyclophilins are involved in several different aspects of disease pathology. This finding is supported, in part, by studies in CypD−/− mice, which showed reduced tubular cell death and a reduction in macrophage infiltration on day 7 UUO. By contrast, CypD−/− mice did not show protection from renal fibrosis on day 7 UUO, although renal fibrosis was significantly reduced on day 12 UUO [27]. Fibroblasts isolated from CypD−/− and wild type UUO kidney showed no difference in terms of mitogen-induced proliferation or TGF-β1 induced upregulation of collagen I and α-SMA mRNA levels [27]. In addition, activation of the p38 MAPK and JNK pathways, which promote renal inflammation and fibrosis in this model [38,39,40], were not affected by CypD deletion [27]. Therefore, the reduction in renal fibrosis on day 12 UUO in CypD−/− mice was attributed to indirect mechanisms, including reduced tubular cell death, reduced inflammation, and protection against loss of peritubular capillaries [27]. GS-642362 treatment also reduced tubular cell death and macrophage infiltration without affecting p38 MAPK or JNK signaling. However, GS-642362 treatment gave a clear-cut inhibition of renal fibrosis on day 7 UUO as demonstrated by a reduction in total kidney collagen content, reduced interstitial deposition of collagen IV, reduced interstitial accumulation of α-SMA+ myofibroblasts, and reduced kidney mRNA levels of collagen IV, α-SMA, and TGF-β1. This suppression of fibrosis was not obviously associated with protection against loss of peritubular capillaries and consequent hypoxia. These findings imply that other cyclophilin family members might also be involved in the pathogenesis of renal fibrosis in the UUO model.

While CypA−/− mice are protected from renal IRI, they do not show protection in the day 7 UUO model in terms of tubular damage, macrophage infiltration, interstitial collagen IV deposition, or upregulation of kidney collagen I or α-SMA mRNA levels [36]. However, we cannot rule out a potential added benefit of blockade of both CypA and CypD using GS-642362 in the UUO model. A recent study in immortalized human HK-2 proximal tubular epithelial cells identified that CypB silencing prevents TGF-β1-induced epithelial to mesenchymal transition-like phenotypic changes [41]. In a day 7 UUO study, CypB−/− mice showed a reduced interstitial macrophage infiltration; however, there was a surprising lack of histologic evidence of renal fibrosis in this model, and the modest upregulation of collagen I and TGF-β1 mRNA levels seen on day 7 UUO in WT mice were unaffected by CypB deletion [41]. Overall, the protective effects of GS-642362 in the UUO model can be attributed to inhibition of CypD, with additional benefits from inhibition of CypB or other members of the cyclophilin family.

The finding that GS-642362 suppressed renal fibrosis in the UUO model is consistent with cyclophilins being involved in the development of fibrosis in non-renal disease models. Non-calcineurin binding cyclophilin inhibitors have been shown to protect mice from viral-induced myocardial fibrosis [42], and from troponin I-induced autoimmune myocardial injury and fibrosis [9]. In addition, a cyclophilin inhibitor reduced liver fibrosis in the CCl_4_ model [43]. However, in contrast to these positive findings, a recent study found that genetic CypD deletion in type 1 diabetic kidney disease, or administration of a cyclophilin inhibitor in type 2 diabetic kidney disease, resulted in exacerbation of kidney injury [44]. Thus, the potential of cyclophilin inhibition as an anti-fibrotic therapy will need careful evaluation.

In conclusion, we have demonstrated that cyclophilin inhibition is effective in suppressing both acute kidney injury and renal fibrosis. These findings warrant further investigation of cyclophilin inhibition, including whether prophylactic cyclophilin inhibition can prevent acute kidney injury in anticipated renal IRI as occurs in patients undergoing coronary artery bypass graft surgery or kidney transplantation.

## 4. Materials and Methods

### 4.1. Materials

Antibodies used in this study were rat anti-mouse F4/80 (Bio-Rad, Gladesville, NSW, Australia); rat anti-mouse Ly6G, rabbit anti-α-SMA, and mouse anti-α-tubulin (Abcam, Melbourne, Victoria, Australia); goat anti-collagen IV (Southern Biotechnology, Birmingham, AL, USA); rabbit anti-phospho-p38 (Thr180/Tyr182) and rabbit anti-phospho-JNK (Thr183/Tyr185) (Cell Signaling, San Diego, CA, USA). Biotinylated goat anti-rabbit IgG was from Zymed-Invitrogen (Carlsbad, CA, USA). The avidin–biotin complex (ABC) kit was from Vector Labs (Burlingame, CA, USA). Alexa Fluor 680 or donkey anti-mouse IRDye 800 secondary antibodies were from Molecular Probes (Thermo Fisher Scientific, Waltham, MA, USA).

### 4.2. Animals

C57BL6/J mice were obtained from the Monash University Animal Research Platform. 

### 4.3. Renal Ischaemia/Reperfusion Injury (IRI)

Bilateral IRI surgery was performed in groups of 10 male C57BL6/J mice using ketamine and xylazine anesthesia as previously described [31]. A heating blanket connected to a rectal thermometer was used to keep body temperature at 37 °C. Following a midline abdominal incision, non-traumatic vascular clamps were used to clamp both renal pedicles for 17 min after which clamps were removed kidney reperfusion checked visually. The abdominal incisions were sutured in two layers, and 0.5 mL saline was provided by subcutaneous injection. At the end of surgery, subcutaneous injection of analgesia agents was provided (0.05 mg/kg Buprenorphine and 4.4 mg/kg Carprofen). GS-642362 was administered via oral gavage to mice in a vehicle consisting of 1:4 (*v*:*v*) PEG300: 50 mM sodium citrate buffer (pH 2.2), and the doses of 10 and 30 mg/kg were selected to cover EC50 and EC90 for CypA in vivo, respectively. Groups of mice received 10 or 30 mg/kg/BID GS-642362 or vehicle only by oral gavage using an 18-gauge flexible plastic tube: the first gavage was given 1 h before surgery, the second gavage between 10 and 12 h after surgery, and the final gavage 1 h before mice were killed at 24 h after reperfusion. Sham operated controls (*n* = 6) underwent the same surgical procedure and analgesia, but renal pedicles were not clamped. Plasma creatinine levels were determined using a Duppon ARL Analyser at the Department of Clinical Biochemistry, Monash Health.

Renal damage was assessed on 2 µm sections of formalin-fixed tissue stained with the periodic-acid-Schiff reagent and hematoxylin. The percentage of tubular cross-sections exhibiting damage in the outer medulla and inner cortex was scored under high power (×400). Damage was defined as one or more of the following: loss of the brush border staining, loss of nuclei, and sloughing of cells into the lumen. Analysis was performed on blinded slides.

### 4.4. Unilateral Ureteric Obstruction (UUO)

Male C57BL6/J mice were anaesthetized with ketamine and xyzaline and a midline incision performed. The left ureter was ligated using two 5-0 silk sutures, and then the incision was sutured in two layers. At the end of surgery, subcutaneous injection of analgesia agents was provided (0.05 mg/kg Buprenorphine and 4.4 mg/kg Carprofen). Groups of mice received 10 or 30 mg/kg/BID GS-642362 or vehicle only by oral gavage, starting 1 h before surgery and continuing twice daily, with the last dose given 1 h before being killed at 7 days after surgery. One group of 8 male mice underwent the same UUO surgical procedure but had no treatment. One group of 6 control male mice did not undergo any experimentation.

### 4.5. Measurement of Peak Plasma Levels of GS-642362

Blood samples were collected into EDTA-containing tubes, centrifuged, and the plasma collected. The plasma sample was deproteinated with two volumes of acetonitrile, containing an internal standard, followed by centrifugation and analysis of the supernatant. Aliquots were analyzed on Thermo TSQ Quantum tandem mass spectrometer connected to an Agilent 1100 Series HPLC with an HTS PAL autosampler by electrospray in SRM mode with positive ionization. Quantification was achieved through the analyte/internal standard peak area ratio as described [26].

### 4.6. Immunohistochemistry

Cell death was visualized by TUNEL staining on 4 μm sections of formalin-fixed tissues using the ApopTag Peroxidase In Situ Apoptosis Detection Kit (Millipore-Chemicon, Ryde, NSW, Australia). The number of TUNEL+ cells were counted under high power (×400) in the outer medulla (IRI model) or in the entire cortex (UUO model).

Immunoperoxidase staining for neutrophils in the IRI model was performed on 5 μm sections of 2% paraformaldehyde-fixed, cryostat tissue sections as previously described [45]. The number of Ly6G+ neutrophils was counted in high power fields (×400) in the outer medulla. Immunostaining for macrophages, α-SMA, collagen IV, and CD31 in the UUO model was performed on 4 µm sections of methylcarn-fixed, paraffin-embedded tissue. The area of interstitial staining for F4/80+ macrophages, α-SMA, collagen IV, and CD31 was determined in the entire cortex (excluding large vessels) under medium power (×200) by image analysis using cellSens software version 1.18 (Olympus Australia, Notting Hill, Victoria, Australia).

### 4.7. Tubular Cell Culture

Primary cultures of renal tubular epithelial cells from C57BL6/J mouse kidney were prepared as previously described [46]. To assay cell death, tubular cells were cultured in 1% fetal calf serum overnight, varying concentrations of GS-642362 was added 1 h before a 24 h incubation with 1mM H_2_O_2_. Cell death was determined using the Cell Death Detection ELISA Kit (Roche, Mannheim, Germany) with results normalized to the DNA content in cell lysates using a Quant-iT DNA Assay Kit (Molecular Probes, Thermo Fisher Scientific, Waltham, MA, USA) and expressed as the ratio of optical density to DNA content. This assay was repeated twice, using replicates of 3, with results pooled.

### 4.8. Real Time Polymerase Chain Reaction (RT-PCR)

Total RNA from frozen kidney samples were prepared using the Ambion RiboPure Kit (Thermo Fisher Scientific, Scoresby, Victoria, Australia) following by random primer-based reverse transcription (SuperScript III First-Strand Synthesis System, Thermo Fisher Scientific, Waltham, MA, USA). PCR reactions were run on the StepOne Real-Time PCR system (Thermo Fisher Scientific, Waltham, MA, USA) using Taqman probes. The primer/probes for NOS2 and α-SMA have been described [38,46]. The other primer/probes were purchased from Thermo Fisher Scientific. The relative amount of mRNA was determined using the comparative Ct (ΔCt) method. All amplicons were normalized against 18S, which was analyzed in the same reaction as an internal control.

### 4.9. Western Blotting

Frozen kidney samples were homogenized in 0.5 mL of lysis buffer as previously described [46]. Samples were then sonicated, incubated at 4 °C for 60 min, centrifuged, and the supernatant heated at 95 °C for 5 min in SDS-PAGE sample buffer. Samples were separated on 4–20% gradient gels and transferred to nitrocellulose membranes. Membranes were incubated with Odyssey Blocking Buffer (LI-COR, Lincoln, NE, USA), incubated with primary antibodies in overnight at 4 °C, washed, incubated with goat anti-rabbit Alexa Fluor 680 or donkey anti-mouse IRDye 800 for 1 h, and scanned with the Odyssey Infrared Image Detecting System (LI-COR). Densitometry analysis was performed by the Gel Pro analyzer program (Media Cybernetics, Rockville, Maryland USA).

### 4.10. Statistics

Data are shown as mean ± SD. Analysis utilized by one-way ANOVA with Tukey’s multiple comparison test using GraphPad Prism (GraphPad Prism 8.0 Software, San Diego, CA, USA).

## Figures and Tables

**Figure 1 ijms-22-00271-f001:**
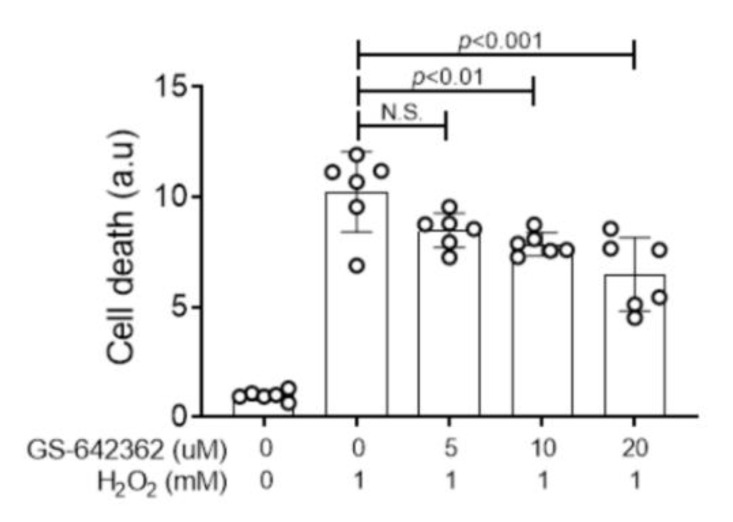
GS-642362 inhibits oxidant-induced cell death. Primary kidney tubular epithelial cells had varying concentrations of GS-642362 added prior to stimulation with 1 mM H_2_O_2_ to induce cell death in primary cultures of tubular epithelial cells. Data shows groups of 6 replicate samples. N.S., not significant.

**Figure 2 ijms-22-00271-f002:**
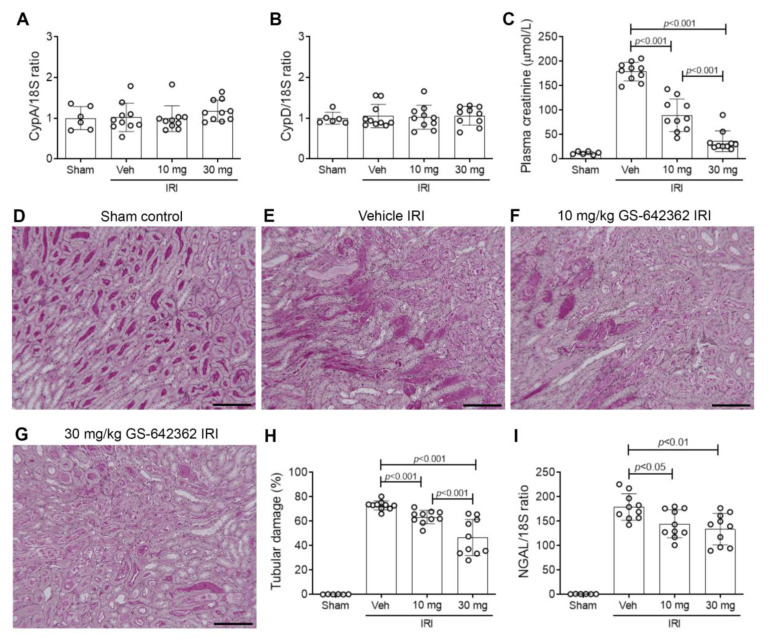
Cyclophilin expression, renal function and tubular damage in renal ischemia/reperfusion injury (IRI). Groups of mice (n = 10) were treated with vehicle, 10 or 30 mg/kg/BID GS-642462 and compared to sham controls (n = 6). RT-PCR for mRNA levels of (**A**) CypA and (**B**) CypD. (**C**) Plasma creatinine levels. (**D**–**G**) Periodic-acid-Schiff-stained kidney sections. (**D**) Sham operated showing normal kidney structure. (**E**) Vehicle-treated IRI showing severe tubular damage in the outer medulla featuring tubular cell loss/sloughing and cast formation. (**F**) IRI treated with 10 mg/kg GS-642362 shows extensive tubular damage. (**G**) IRI treated with 30 mg/kg GS-642362 shows lesser, but still significant, tubular damage. Bar = 200 µm. (**H**) Graph of tubular damage. (**I**) RT-PCR for mRNA levels of the tubular damage marker, NGAL/LCN2.

**Figure 3 ijms-22-00271-f003:**
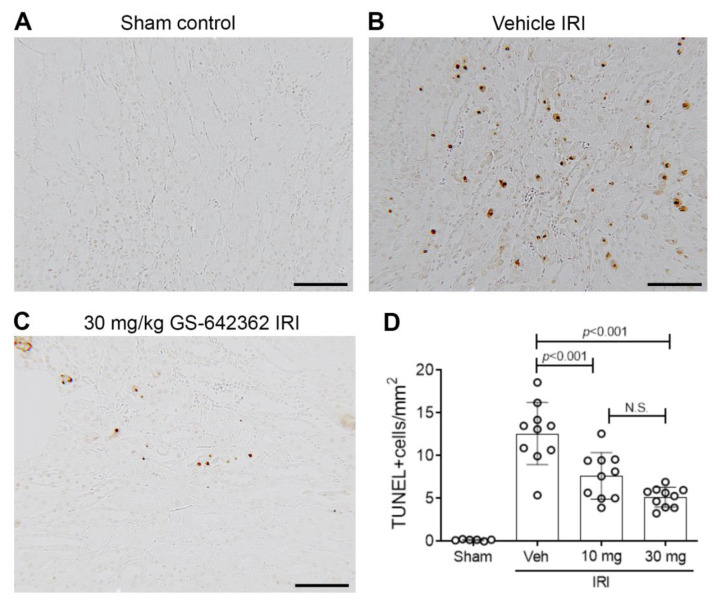
Cell death in renal ischemia/reperfusion injury (IRI). Groups of mice (n = 10) were treated with vehicle, 10 or 30 mg/kg/BID GS-642462 and compared to sham controls (n = 6). (**A**–**C**) TUNEL staining. (**A**) Sham control lacks TUNEL+ cells. (**B**) Vehicle-treated IRI showing many TUNEL+ tubular cells in the outer medulla. (**C**) IRI treated with 30 mg/kg GS-642362 shows a marked reduction in the number of TUNEL+ cells. Bar = 100 µm. (**D**) Graph of TUNEL+ cells.

**Figure 4 ijms-22-00271-f004:**
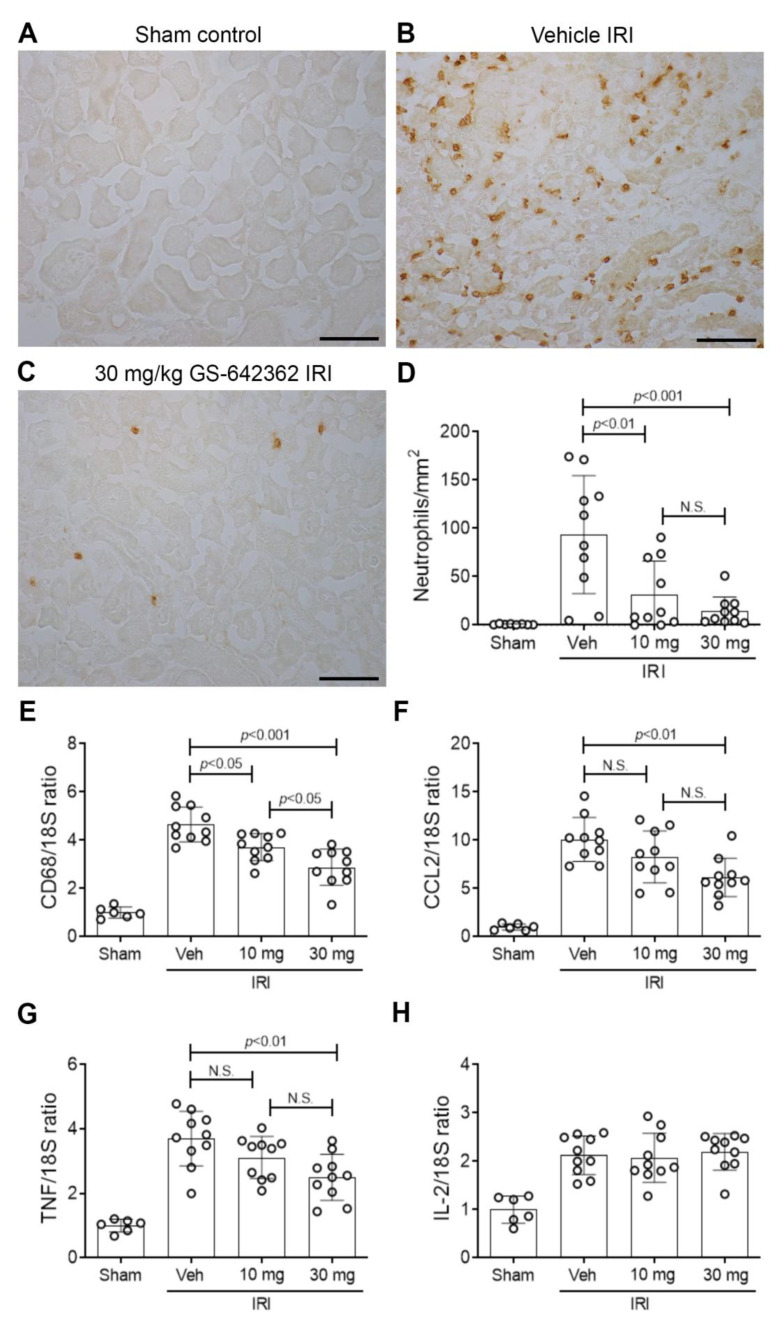
Neutrophil and macrophage infiltration in renal ischemia/reperfusion injury (IRI). Groups of mice (n = 10) were treated with vehicle, 10 or 30 mg/kg/BID GS-642462 and compared to sham controls (n = 6). (**A**–**C**) Immunostaining of Ly6G+ peroxidases. (**A**) Sham control lacks neutrophils. (**B**) Vehicle-treated IRI shows many infiltrating neutrophils in an area of tubular damage. (**C**) IRI treated with 30 mg/kg GS-642362 shows a dramatic reduction in neutrophil infiltration. Bar = 100 µm. (**D**) Graph of Ly6G+ neutrophils. RT-PCR analysis of mRNA levels for (**E**) CD68, (**F**) CCL2, (**G**) TNF, and (**H**) IL-2. N.S., not significant.

**Figure 5 ijms-22-00271-f005:**
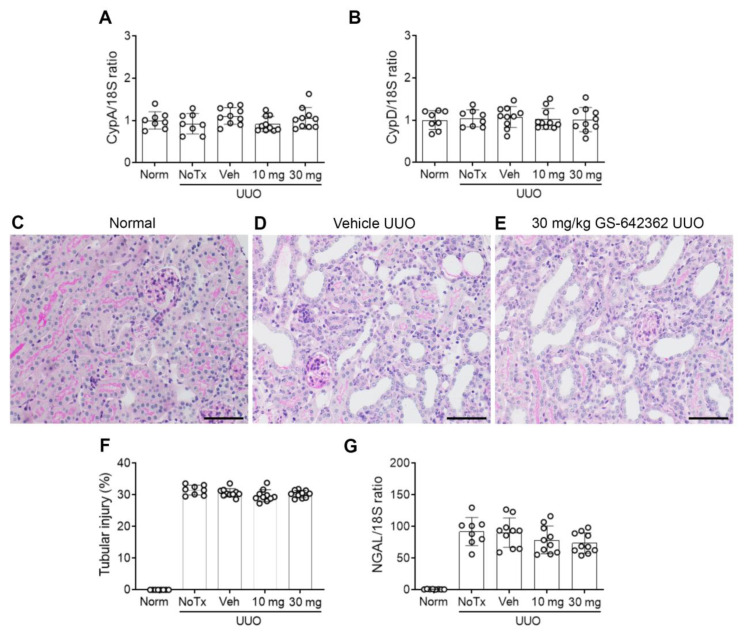
Cyclophilin expression and tubular damage in the day 7 unilateral ureteric obstruction (UUO) model (n = 8 to 10/group). RT-PCR for mRNA levels of (**A**) CypA and (**B**) CypD. (**C**–**E**) Periodic-acid-Schiff-stained kidney sections. (**C**) Normal mouse kidney. (**D**) Vehicle-treated UUO showing marked tubular dilation. (**E**) UUO treated with 30 mg/kg GS-642362 also shows marked tubular dilation. Bar = 200 µm. (**F**) Graph of tubular damage. (**G**) RT-PCR for mRNA levels of the tubular damage marker, NGAL/LCN2.

**Figure 6 ijms-22-00271-f006:**
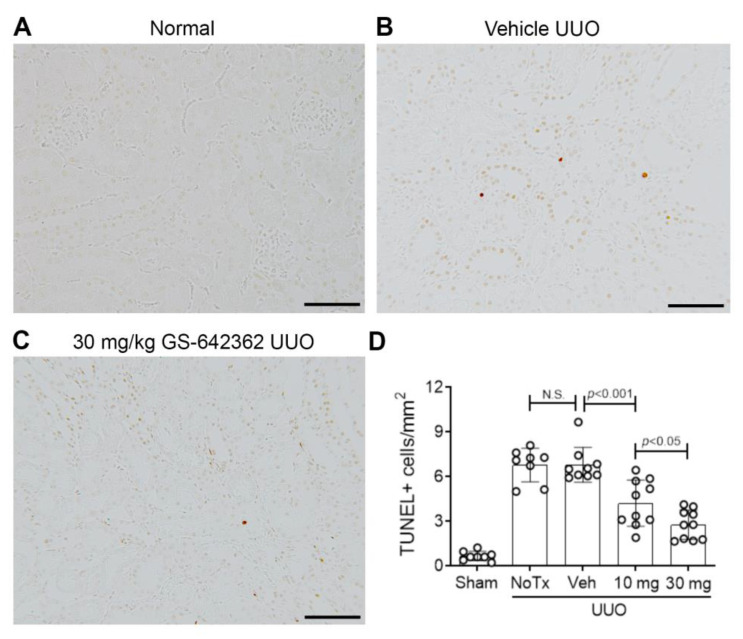
Cell death in the day 7 unilateral ureteric obstruction (UUO) model (n = 8 to 10/group). (**A**–**C**) TUNEL staining. (**A**) Normal mouse kidney. (**B**) Vehicle-treated UUO showing small numbers of TUNEL+ tubular cells in the cortex. (**C**) UUO treated with 30 mg/kg GS-642362 shows a clear reduction in the number of TUNEL+ cells. Bar = 100 µm. (**D**) Graph of TUNEL+ cells. N.S., not significant.

**Figure 7 ijms-22-00271-f007:**
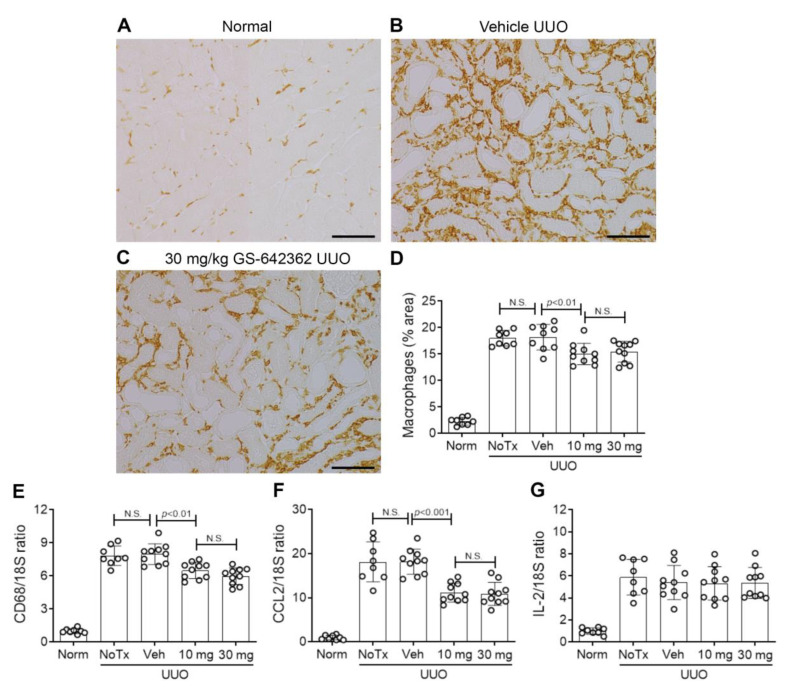
Macrophage infiltration in the day 7 unilateral ureteric obstruction (UUO) model (n = 8 to 10/group). (**A**–**D**) Immunostaining for F4/80+ macrophages. (**A**) Normal mouse kidney showing resident F4/80+ macrophages. (**B**) Vehicle-treated UUO showing a significant increase in macrophages in the cortex. (**C**) UUO treated with 30 mg/kg GS-642362 shows a partial in macrophage infiltration. Bar = 100 µm. (**D**) Graph of the area of F4/80 macrophage staining. RT-PCR for mRNA levels of (**E**) CD68, (**F**) CCL2, and (**G**) IL-2. N.S., not significant.

**Figure 8 ijms-22-00271-f008:**
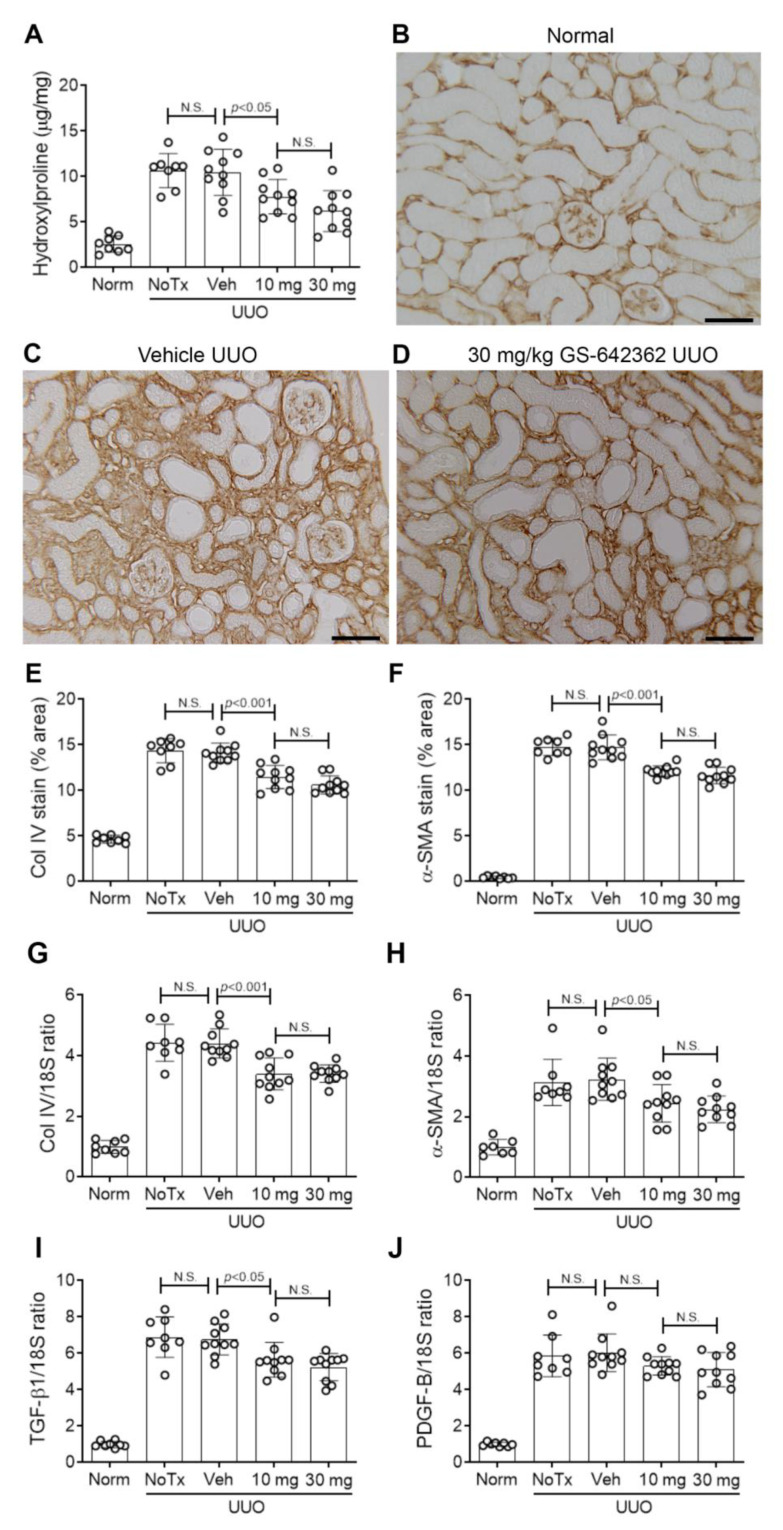
Renal fibrosis in the day 7 unilateral ureteric obstruction (UUO) model (n = 8 to 10/group). (**A**) Hydroxyproline assay of total collagen content. (**B**–**D**) Immunostaining for collagen IV. (**B**) Normal kidney shows fine staining of collagen IV in the glomerular and tubular basement membranes. (**C**) Vehicle-treated UUO shows a substantial increase in interstitial collagen IV staining. (**D**) UUO treated with 30 mg/kg GS-642362 shows a partial in interstitial collagen IV staining. Bar = 100 µm. (**E**) Graph of the area of interstitial collagen IV staining. (**F)** Graph of the area of interstitial staining for α-SMA. RT-PCR for mRNA levels of (**G**) collagen IV, (**H**) α-SMA/Acta2, (**I**) TGF-β1, and **(J**) PDGF-B. N.S., not significant.

**Figure 9 ijms-22-00271-f009:**
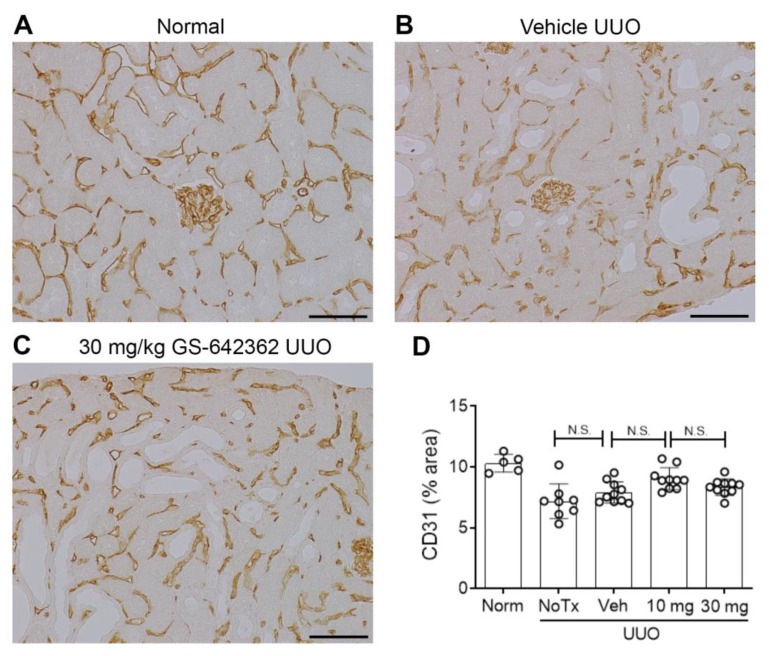
Peritubular capillary loss in the day 7 unilateral ureteric obstruction (UUO) model (n = 8 to 10/group). (**A**–**C**) Immunostaining for CD31+ endothelial cells. (**A**) Normal kidney shows a dense network of peritubular capillaries and glomerular capillary tufts. (**B**) Vehicle-treated UUO shows a partial loss of peritubular capillaries. (**C**) UUO treated with 30 mg/kg GS-642362 also shows a partial loss of peritubular capillaries. Bar = 100 µm. (**D**) Graph of the area of interstitial CD31 staining. N.S., not significant.

**Figure 10 ijms-22-00271-f010:**
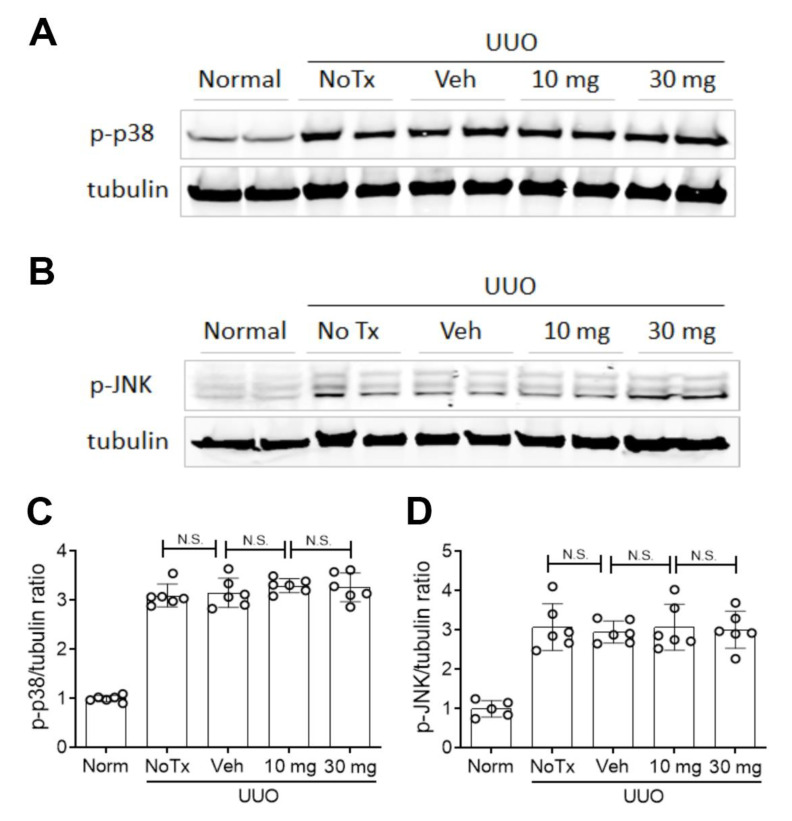
Stress-activated protein kinases in the day 7 unilateral ureteric obstruction (UUO) model (n = 8 to 10/group). Western blots for (**A**) phospho-p38 MAPK and (**B**) phospho-JNK, with a tubulin loading control. Graphs show quantification for (**C**) phospho-p38 MAPK and (**D**) phospho-JNK. N.S., not significant.

## Data Availability

Data is available on request.

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
