# Peer review of "Cyclophilin Inhibition Protects Against Experimental Acute Kidney Injury and Renal Interstitial Fibrosis"

_ijms, 2020, doi:10.3390/ijms22010271_

Round 1
Reviewer 1 Report
Comments to the reviewer
Their recent study has reported that cyclophilin A (CypA) deficiency mice had less kidney damage following renal ischemia/reperfusion injury (IRI) compared with wild-type mice, while CypA gene knockout did not prevent renal fibrosis formation in unilateral ureteric obstruction (UUO) mode. In this study, the authors would like to extend their finding and further investigated the effect of a cyclophilin inhibitor, GS-642263, on alleviation of renal IRI and renal fibrosis. They demonstrated a reduced kidney injury following GS-642263 treatment in the renal IRI model and a protective effect of this chemical compound on renal fibrosis in the UUO mode.
- The most criticized point of this study is the lack of the mechanistic investigation regarding how this chemical specifically blocks cyclophilins to prevent kidney injury. Which pathway is implicated in the GS-642263-mediated inhibition of kidney injury? Is any specific molecule interacted with cyclophilins altered following GS-642263 administration?
- As the description of the authors, this specific cyclophilin inhibitor does not affect immunity like cyclosporine (CsA). Calcineurin/cyclophilin A binding is critical for the cyclophilin A/calcineurin/nuclear factor of T cell activator (NFAT). It would be more convincing if the expressions of these molecules can be quantified in this study to verify the specificity of this compound. Which cyclophilin expression (CypA, CypB or CypD) is suppressed by GS-642263 as this compound is a specific cyclophilin inhibitor?
- To verify the specific effect of GS-642263 on prevention of kidney injury and kidney fibrosis, it would be better to compare the effect of other cyclophilin inhibitors such as debio 025 or NIM811 with this compound. Further, does CsA have different effects in these two modes? As GS-642263 is a specific cyclophilin inhibitor without affecting calcineurin/NFAT, it would be better to demonstrate that the cyclophilin-dependent but NFAT-independent pathway is crucial for renal proetection.
- In their previous study, CypA deficiency did not affect tubulointerstitial fibrosis in UUO mice but this cyclophilin inhibitor reduced tubulointerstitial fibrosis in this study. The authors speculated that CypD suppression may contribute to the protective effect of GS-642263. However, it would be better to show the evidence, CypD inhibition. It seems this compound is not very specific for specific cyclophilin inhibition. CypA, CypB and CypD have some opposite effects in different biofunctions. The authors need to address the issue which cyclophilin is the main effector of GS-642263.
Author Response
Reviewer #1
Their recent study has reported that cyclophilin A (CypA) deficiency mice had less kidney damage following renal ischemia/reperfusion injury (IRI) compared with wild-type mice, while CypA gene knockout did not prevent renal fibrosis formation in unilateral ureteric obstruction (UUO) model. In this study, the authors would like to extend their finding and further investigated the effect of a cyclophilin inhibitor, GS-642263, on alleviation of renal IRI and renal fibrosis. They demonstrated a reduced kidney injury following GS-642263 treatment in the renal IRI model and a protective effect of this chemical compound on renal fibrosis in the UUO model.
- The most criticized point of this study is the lack of the mechanistic investigation regarding how this chemical specifically blocks cyclophilins to prevent kidney injury. Which pathway is implicated in the GS-642263-mediated inhibition of kidney injury? Is any specific molecule interacted with cyclophilins altered following GS-642263 administration?
Reply: it is technically very difficult to measure direct molecular interactions of GS-642362 with individual molecules in kidney tissue. However, GS-642632 has been shown to bind to CypA using a time-resolved fluorescence resonance energy transfer probe displacement assay in vitro [ref 26], validating this compound as a cyclophilin inhibitor. GS-642362 can also bind and inhibit CypD as demonstrated by the drug-based inhibition of H2O2-induced tubular cell death (Fig 1); a response that operates via CypD and not via CypA (refs 26 and 35). Thus, we can confidently conclude that GS-642362 inhibits both CypA and CypD functions in disease pathogenesis. In addition, we have confirmed that GS-642362 did not inhibit the cyclosporin A target, NFAT, since the up-regulation of IL-2 mRNA levels in both IRI and UUO models was not affected by drug treatment (new data in Figs 4H and 7G in the revised manuscript). Therefore, we can conclude from our study that GS-642362 inhibited CypD-mediated tubular cell death in both IRI and UUO models, that GS-642362 inhibited CypA-mediated neutrophil infiltration in the IRI model, and that these effects were independent of NFAT activation.
- As the description of the authors, this specific cyclophilin inhibitor does not affect immunity like cyclosporine (CsA). Calcineurin/cyclophilin A binding is critical for the cyclophilin A/calcineurin/nuclear factor of T cell activator (NFAT). It would be more convincing if the expressions of these molecules can be quantified in this study to verify the specificity of this compound. Which cyclophilin expression (CypA, CypB or CypD) is suppressed by GS-642263 as this compound is a specific cyclophilin inhibitor?
Reply: GS-642362, like other cyclophilin inhibitors, operate by direct interaction with cyclophilin proteins without affecting cyclophilin expression levels. Cyclophilins A and D are highly expressed in the kidney. The induction of IRI or UUO models did not change CypA or CypD mRNA levels (new data in Figures 2A, B and 5A, B in the revised manuscript). As expected, GS-642632 treatment did not affect CypA or CypD mRNA levels. NFAT acts directly to transcribe the IL-2 gene. Our new data shows that GS-642362 did not affect NFAT activation based on the lack of effect upon the up-regulation of IL-2 mRNA levels in both the IRI and UUO models (Figures 4H and 7G in the revised manuscript).
- To verify the specific effect of GS-642263 on prevention of kidney injury and kidney fibrosis, it would be better to compare the effect of other cyclophilin inhibitors such as debio 025 or NIM811 with this compound. Further, does CsA have different effects in these two models? As GS-642263 is a specific cyclophilin inhibitor without affecting calcineurin/NFAT, it would be better to demonstrate that the cyclophilin-dependent but NFAT-independent pathway is crucial for renal protection.
Reply: as described above, we have shown that the up-regulation of the NFAT transcribed gene, IL-2, is not affected by GS-642362 treatment in the IRI and UUO models. Cyclosporin A can both prevent and exacerbate acute renal IRI depending upon the dose and the severity of injury; however, chronic administration of cyclosporin A induces renal interstitial fibrosis in otherwise healthy animals. This point has been added to the Introduction (lines 56-60) in the revised manuscript. Indeed, this well-recognised toxicity has led to very careful monitoring of plasma cyclosporin A levels in transplant patients, and considerable efforts to develop cyclosporin A treatment sparing protocols. Neither NIM811 or debio-025 have been tested in renal IRI, and to perform a head-to-head comparison of GS-642362 with these other cyclophilin inhibitors in the two models is a considerable undertaking that is beyond the scope of the current study.
- In their previous study, CypA deficiency did not affect tubulointerstitial fibrosis in UUO mice but this cyclophilin inhibitor reduced tubulointerstitial fibrosis in this study. The authors speculated that CypD suppression may contribute to the protective effect of GS-642263. However, it would be better to show the evidence, CypD inhibition. It seems this compound is not very specific for specific cyclophilin inhibition. CypA, CypB and CypD have some opposite effects in different biofunctions. The authors need to address the issue which cyclophilin is the main effector of GS-642263.
Reply: this is a good point and highlights many of the unknown aspects of how the different cyclophilin family members act in different aspects of disease pathology. While we have not shown direct inhibition of CypD/mPTP function by GS-642362, the demonstration that GS-642362 inhibited tubular cell death in both IRI and UUO models, as well as in H2O2-induced cell death in cultured tubular cells, is entirely consistent with inhibition of the CypD/mPTP interaction. Similarly, while GS-642362 has been shown to bind to and inhibit CypA in vitro, we have not directly demonstrated this in vivo. However, the dramatic reduction in neutrophil infiltration in the IRI model is entirely consistent with GS-642362 inhibition of CypA. As we indicated in the Discussion, we think that protection in IRI is due to combined inhibition of CypA and CypD, whereas protection in UUO is mostly due to inhibition of CypD, but with possibly a contribution from CypB inhibition. To demonstrate this in precise detail would require studies using all 3 KO mice with and without drug treatment, which is beyond the scope of these studies.
Reviewer 2 Report
Summary:
The authors tested the cyclophilin inhibitor, GS-642362, in models of acute kidney injury and renal fibrosis. GS-642362 was found to improve renal function, reduce cell death and reduce neutrophil infiltration following bilateral ischemia reperfusion injury (IRI). In a model of unilateral ureteric obstruction (UUO), GS-642362 reduced cell death, reduced macrophage infiltration and reduced fibrotic development, albeit to a less degree than in the model of IRI. Previous studies have focused on genetic depletion of cyclophilin A (CypA) and cyclophilin D (CypD) and there have been relatively few studies that have taken a pharmacological inhibition approach of these molecules. In this regard, the study makes an important contribution by showing that pharmacological blockade has a key effect on acute and chronic kidney injury. Moreover, this suggests the potential for the translational use for these compounds. Notwithstanding the above comments, the current study could be greatly improved by addressing some of the concerns related below.
Strengths:
- This study complements previous work performed by genetic depletion of CypA and CypD
- The authors explore two different models of kidney injury (ischemia reperfusion injury and unilateral ureteric obstruction). This is important as IRI has an early phase of inflammation following by repair and fibrosis and UUO is a classic way kidney investigators focus on the chronic fibrotic progression of kidney disease.
- The authors use two different concentrations of the inhibitor allowing for a dose-response analysis
Weaknesses:
- The study would be strengthened by a more complete description of treatment protocol for administration of inhibitor.
- The study would be strengthened by reporting the plasma levels of the inhibitor. This is stated in the manuscript, but how they are measured is unclear. This is particularly important because of the acute administration and the short timeline prior to sacrifice, especially for IRI studies.
- The work would be greatly strengthened by the inclusion of experiments that show reduction of CypA and CypD does occur based on the administration of the inhibitor. This is an important experiment as it would help to define the mechanisms responsible for the observations on the progression of kidney injury.
- The work would be strengthened by relying on more than one measure of injury. For example, tubular injury was represented by PAS staining. This could be strengthened by showing expression of injury markers such as NGAL and KIM1. In addition to measures of injury, investigators should couple their measures of genes involved in inflammation such as pro-inflammatory cytokines.
- The measures of injury should be similar in both groups. The authors only show PAS for IRI studies, not UUO (state there was no injury, should still show). The authors only show F4/80 staining and fibrotic gene expression in the UUO experiments and rely solely on CD68 gene expression to represent macrophage infiltration following IRI.
Conclusions: Study could be strengthened by more extensive measures of inflammation and fibrosis and by utilizing the same measurements in both models. Critically, the authors should perform studies on expression of CypA and CypD to study the mechanism responsible.
Author Response
Reviewer #2
Summary:
The authors tested the cyclophilin inhibitor, GS-642362, in models of acute kidney injury and renal fibrosis. GS-642362 was found to improve renal function, reduce cell death and reduce neutrophil infiltration following bilateral ischemia reperfusion injury (IRI). In a model of unilateral ureteric obstruction (UUO), GS-642362 reduced cell death, reduced macrophage infiltration and reduced fibrotic development, albeit to a less degree than in the model of IRI. Previous studies have focused on genetic depletion of cyclophilin A (CypA) and cyclophilin D (CypD) and there have been relatively few studies that have taken a pharmacological inhibition approach of these molecules. In this regard, the study makes an important contribution by showing that pharmacological blockade has a key effect on acute and chronic kidney injury. Moreover, this suggests the potential for the translational use for these compounds. Notwithstanding the above comments, the current study could be greatly improved by addressing some of the concerns related below.
Strengths:
- This study complements previous work performed by genetic depletion of CypA and CypD
- The authors explore two different models of kidney injury (ischemia reperfusion injury and unilateral ureteric obstruction). This is important as IRI has an early phase of inflammation following by repair and fibrosis and UUO is a classic way kidney investigators focus on the chronic fibrotic progression of kidney disease.
- The authors use two different concentrations of the inhibitor allowing for a dose-response analysis
Weaknesses:
- The study would be strengthened by a more complete description of treatment protocol for administration of inhibitor.
Reply: we have provided a more detailed description of the treatment protocol in the Methods sections 4.3 and 4.4. covering the IRI and UUO models, which includes moving the description of GS-642362 formulation into section 4.3.
- The study would be strengthened by reporting the plasma levels of the inhibitor. This is stated in the manuscript, but how they are measured is unclear. This is particularly important because of the acute administration and the short timeline prior to sacrifice, especially for IRI studies.
Reply: the method for measuring plasma drug levels has been included as the new section 4.5 in the Methods section.
- The work would be greatly strengthened by the inclusion of experiments that show reduction of CypA and CypD does occur based on the administration of the inhibitor. This is an important experiment as it would help to define the mechanisms responsible for the observations on the progression of kidney injury.
Reply: GS-642362, like other cyclophilin inhibitors, operate by direct interaction with cyclophilin proteins without affecting cyclophilin expression levels. We have included new data (see Figs 2A, B and 5A, B in the revised manuscript) showing that the induction of IRI or UUO models did not change CypA or CypD mRNA levels, and that GS-642632 treatment did not affect CypA or CypD mRNA levels in either model.
- The work would be strengthened by relying on more than one measure of injury. For example, tubular injury was represented by PAS staining. This could be strengthened by showing expression of injury markers such as NGAL and KIM1. In addition to measures of injury, investigators should couple their measures of genes involved in inflammation such as pro-inflammatory cytokines.
Reply: we have included new data for the tubular marker, NGAL, in both models (Fig 2I and 5F in the revised manuscript). In the IRI model, GS-642362 significantly reduced NGAL mRNA levels, consistent with the reduction in tubular damage seen by PAS staining. In the UUO model, the upregulation of NGAL mRNA levels was not affected, consistent with the lack of effect on tubular damage seen on PAS staining – as expected since this injury is due to mechanical stretch caused by fluid retention following ureteric ligation. We have also included new data on inflammatory markers, so that both IRI and UUO models show data for CCL2 and IL-2 mRNA levels (see Figures 4 and 7 in the revised manuscript).
- The measures of injury should be similar in both groups. The authors only show PAS for IRI studies, not UUO (state there was no injury, should still show). The authors only show F4/80 staining and fibrotic gene expression in the UUO experiments and rely solely on CD68 gene expression to represent macrophage infiltration following IRI.
Reply: we have included images of the PAS staining in the UUO model, in which there was no difference in tubular damage, together with NGAL expression also showing no difference in tubular damage (see new Fig 5 in the revised manuscript). We have not included images of the F4/80 macrophage staining in the IRI model since this used paraformaldehyde-fixed tissue which is sub-optimal for immunostaining for the F4/80 antigen compared to the methylcarn-fixed tissue used in the UUO model (methylcarn-fixed tissue was not collected in the IRI model).
Conclusions: Study could be strengthened by more extensive measures of inflammation and fibrosis and by utilizing the same measurements in both models. Critically, the authors should perform studies on expression of CypA and CypD to study the mechanism responsible.
Reply: we have provided new data covering expression of CypA and CypD mRNA levels in both disease models, the NGAL marker of tubular damage in both models, and data on the inflammation markers CCL2 and IL-2 in both models (details above). We have not assessed fibrosis in the acute IRI study as fibrosis does not develop in this model until day 14.
Round 2
Reviewer 1 Report
No more comments.
Author Response
No further comments.
Reviewer 2 Report
The authors addressed the majority of comments in the revised version. I have minor comments to still be address. 1) No title for results section on line 79 2) Figure 2D and 2E are labelled wrong on line 89 3) Quantification of tubular damage should be displayed in Figure 5 4) In Figure 6, I would recommend showing representative images of sham/normal mice as opposed to UUO with no treatment to be consistent.Author Response
1) No title for results section on line 79
Reply: This has been corrected.
2) Figure 2D and 2E are labelled wrong on line 89
Reply: This has been corrected.
3) Quantification of tubular damage should be displayed in Figure 5
Reply: This has been added as Figure 5G
4) In Figure 6, I would recommend showing representative images of sham/normal mice as opposed to UUO with no treatment to be consistent.
Reply: As requested, we have replaced the no treatment UUO image of TUNEL staining in Figure 6A with an image of TUNEL staining of normal kidney.